# Research on the Coupling and Coordination of Systems of Citizenization, Regional Economy, and Public Service in China from the Perspective of Sustainable Development

**Mingzhu Qi [1,\*], Peng Miao [2], Ya Wang [2] and Yuge Song [3]**

[1] Population and Economic Research Center, School of Labor Economics, Capital University of Economics and Business, Beijing 100070, China
[2] School of Labor Economics, Capital University of Economics and Business, Beijing 100070, China
[3] UCLA College of Letters and Science, University of California, Los Angeles, Los Angeles, CA 90095, USA
[\*] Correspondence: qimzh@cueb.edu.cn; Tel.: +86-13910108216

**Abstract:** Based on the perspective of sustainable development, this paper introduces the circular economy dimension into the economic system, the ecological environmental protection dimension into the public service system, and constructs index evaluation systems for citizenization, regional economy, and public service, respectively. Then, the paper establishes a coupling coordination model to study the mutual relationship between citizenization, regional economy, and public service at provincial levels, so as to promote the sustainable development of population, economic society, and resource environment. The main research conclusions are as follows: (1) the level of citizenization is generally low, and the dimensions of income and social security are weak; (2) the development trend of circular economy in the eastern regions is better than that in other regions; (3) the "Citizenization, Regional Economy, Public Service" System has an overall low coupling and coordination degree and it shows a trend of "high in the east and low in the west"; (4) the contradictory characteristics between the systems vary greatly by province. Most of the provinces' systems of the regional economy and public service lag behind their citizenization and some developed provinces have a lag in their citizenization systems. In order to promote a high-quality coordinated development between the systems of citizenization, regional economy, and public service, a sustainable development path that adapts to local conditions must be sought. It should focus on improving the social security level of rural-to-urban migrants, establishing a circular economy, and strengthening the construction of an ecological environment.

**Keywords:** rural-to-urban migrants; sustainable development; coupling coordination degree; citizenization; circular economy

## 1. Introduction

The transfer of agricultural population to cities occurs all over the world [1]. However, due to the particular household registration system in China, the issue of granting city residency to rural-to-urban migrants is unique (in the following parts, to make the presentation concise, we use "citizenization" to substitute for "granting city residency to rural-to-urban migrants"). Although they have moved to cities, they cannot obtain the household registration residency (*hukou*) in the inflow locales, so they have not been able to enjoy the same essential public services as the urban residents in those locales and are in the situation of "semi-citizenization". For example, rural-to-urban migrants without local *hukou* usually cannot work in state-owned enterprises and public institutions that tend to provide higher levels of various social securities. In some megacities, people cannot buy a house or a car without a local *hukou* and their children cannot attend local public schools. The goal of citizenization is to allow the rural-to-urban migrants to have the same treatments as the urban residents with *hukou*. By the end of 2021, the amount of

rural-to-urban migrants exceeded 250 million. It not only requires accelerating the progress of granting migrants the same treatments but also adapting the citizenization progress to the capability of public services and the level of regional economic development in the inflow locales.

The core of the new-type urbanization is the urbanization of people through the citizenization of rural-to-urban migrants, while promoting the equalization of public services is a key measure of citizenization. At the same time, the new normal of China's economy requires the transformation of the economic development mode from the scale–speed type to the quality–efficiency type. Therefore, the circular economy has become an important choice for the sustainable development of the regional economy. The level of regional economic development affects the supply capacity of public services, thereby restricting the process of citizenization. Moreover, while moderate citizenization puts forward new requirements for the public service system, it can also effectively promote regional economic development. Therefore, a coordinated development between the three systems of citizenization, regional economy, and public service is critical. However, local governments do not realize this importance in the current promotion of citizenization. They lack attention to the coordinated development among the three systems and do not unite the "citizenization process" with the economy and the public service systems. The mismatch of the three systems brings additional difficulties to rural-to-urban migrants and may lead to new urban-rural social conflicts, which are not conducive to regional sustainable development.

Under such background, this paper constructs a coupling coordination model of the "Citizenization, Regional Economy, Public Service" system and analyzes the coupling coordination feature of the three systems, aiming at realizing a high-quality citizenization and its sustainable coordinated development with regional economy and public services.

The remainder of the paper is arranged as follows. In Section 2, we present a review of the literature. In Section 3, an index system is constructed and the research methods are listed. Section 4 presents a coupling and coordination analysis between the three systems. Finally, Section 5 states the conclusion and the discussion of this paper, emphasizing the paper's contribution and some proposals for future research.

## 2. Review of the Literature

"Citizenization" referred to the process in which rural-to-urban migrants obtain the same legal status and social rights as urban residents, mainly involving urban household registration (*hukou*) and equal treatment in education, labor employment, social security, public services, etc., [2–4]. Rural migration in other countries was generally characterized by farmers moving to cities, being absorbed by modern industrial sectors, and becoming urban residents with access to all urban services. There was no semi-citizenization stage. The unique citizenization problem in China had not yet attracted extensive attention from foreign scholars, and there was a lack of research on the coupling and coordination relationship between systems of citizenization, regional economy, and public service. The relevant literature mainly studied the binary or multivariate relationship between urbanization and economic development, public services, ecological energy, etc. Northam (1975) proposed a positive linear relationship between economic development and urbanization [5]. Zweig (2001) studied the economic imbalance between rural and urban China and the consequent labor migration [6]. Lall et al., (2006) extensively assessed the migration over the past five decades from rural to urban areas in developing countries, including China [7]. They argued that migration was generally beneficial to economic growth, while policy restrictions on migration were not desirable. Economic growth could provide essential financial support and better public services, including infrastructure construction, and could promote better solutions to citizenization or ecological problems, while improving social and environmental quality could also provide economic development and offer a better foundation and more possibilities in turn [8–10]. Colombier (2011) analyzed a sample in Switzerland, showing that spending on public transport infrastructure is beneficial to economic development [11]. Ali (2013) took Pakistan as a sample, and the results also verified that public

service expenditure positively affects economic growth [12]. Bakken (2015) argued that although the migration of the rural population to cities had revitalized China's economy, it also posed a threat to social order [13]. Antonelli et al., (2019) analyzed 22 OECD countries and found that the efficiency of public service expenditure significantly impacted economic development [14]. Bakirtas et al., (2018) studied the relationship between urbanization, economic growth, and energy consumption in emerging market countries (Colombia, India, Indonesia, Kenya, Malaysia, and Mexico) from 1971 to 2014 and found that the three were interdependent [15]. Chu (2022) and others studied population urbanization, economic urbanization, social urbanization, ecological environment urbanization, and their coupling and coordinated development in Russia from 2005 to 2020 [16]. Chinese scholars had also verified the positive interaction between public services, urbanization, and regional economy [17,18].

Although domestic scholars had paid attention to the issue of citizenization, most of the relevant studies started from a micro perspective, studying the microscopic influencing factors of promoting citizenization and ignoring the macroscopic aspects of the citizenization system. Currently, most of the Chinese scholars' relevant research on the systems of citizenization, regional economy, and public service only focus on the relationship between two systems. First, in terms of the relationship between citizenization and regional economy, citizenization could bring economic benefits to the region [19,20] and a good economy could provide financial support for citizenship [21–23]. However, the process of citizenization in some megacities with high economic levels was hindered by restrictions on population size [24] and the threshold for being granted urban residency was very high [25]. Second, in terms of the relationship between citizenization and public services, having the access to essential public services in a city was a key realization of citizenization [21,22], while expanding the supply of public services could significantly increase rural-to-urban migrants' willingness to become urban residents [26]. However, the current contradiction remains in that, under the household registration system, the supply capacity of public services was not equal to the actual supply level; especially, the point-based household registration system in megacities still had a high threshold, so rural-to-urban migrants could not fully enjoy the public services of the inflow locales [27]. Third, regarding the relationship between public services and regional economic development, the former was the primary social condition for the latter and promoted sustained economic growth to a certain extent. Meanwhile, regional economic growth provided financial support for expanding the supply of public services, which enabled the continuous enhancement of supply capacity [28,29]. In addition, few studies considered circular economy and environmental protection when discussing the relationship between the three systems. The citizenization of the rural-to-urban migrants could increase the population in the city, which would put pressure on resources and the environment and increase energy consumption, hindering the sustainable development of cities [30,31]. Focusing on green development indicators such as circular economy would help the three systems to achieve a higher-quality coordinated development [32].

To sum up, there are still two deficiencies in the relevant literature. First, there is insufficient attention to the coordinated development of the systems of citizenization, regional economy, and public service. Second, in terms of measurement methods, the index system does not involve sustainable development indicators of ecology, resources, and environment, which are essential for achieving the goal of high-quality economic development. For example, the circular economy is integral to regional sustainable development. However, existing research rarely incorporates circular economy indicators in the measurement of regional economic systems, and the public service system lacks environmental protection indicators and does not reflect the concept of sustainable development.

Given the above limitations, first of all, to fully reflect the requirements of sustainable development in the selection of indicators, this paper introduces elements of circular economy into the economic system and elements of environmental protection into the public service system. Secondly, we put the systems of citizenization, regional economy, and public

service under the same framework, discuss the coupling coordination degree between the three systems and analyze their coordination relationship, as well as their regional differences. Finally, policy recommendations to promote the realization of sustainable development between systems are proposed.

## 3. Research Methods

### 3.1. Data Sources

In this paper, the data for the citizenization index system of rural-to-urban migrants come from the China Migrants Dynamic Survey (CMDS). These surveys have been carried out annually since 2009; they cover 31 provinces/autonomous regions/municipalities (hereinafter referred to as "provinces").

Data were released most recently in 2018, which are used in this paper. The CMDS sample is composed of floating population members aged 15 and above who have lived in their inflow locales for more than one month but do not have household registration (*hukou*) in those locales. Two categories of inflow and outflow types are covered in the survey: urban and rural. The total sample size of CMDS 2018 was 152,001, and of that number, 103,329 had agricultural *hukou*, from whom we selected migrants whose current residence was in an urban locale. This yielded a sample of 70,001. The data for the systems of regional economy and public service come from the 2018 China Statistical Yearbook, China Health Statistical Yearbook, China Urban Statistical Yearbook, China Population, Employment Statistical Yearbook, and the National Bureau of Statistics website. Further, to explore the development trend of the circular economy in recent years, we supplement the circular economy data from 2016 to 2020, which are from the 2016-2020 China Energy Statistics Yearbook. See Table 1 for more details.

**Table 1.** Data Sources.

| System | Dimension | Data Sources |
| --- | --- | --- |
| Citizenization System | Employment | China Migrants Dynamic Survey in 2018 |
| | Income | China Migrants Dynamic Survey in 2018 |
| | Social security | China Migrants Dynamic Survey in 2018 |
| Regional Economy System | Basic economy | China Statistical Yearbook in 2018 |
| | Circular economy | China Energy Statistics Yearbook in 2018 |
| Public Service System | Science, education, culture, and health | China Health Statistical Yearbook in 2018 |
| | Infrastructure | The National Bureau of Statistics website and China Urban Statistical Yearbook in 2018 |
| | Insurance payment | China Population, Employment Statistical Yearbook in 2018 |
| Supplementary Note | Ecological environment | China Urban Statistical Yearbook in 2018 |
| | Development trend of circular economy | China Energy Statistics Yearbook in 2016–2020 |

### 3.2. Index System Construction and Weight Determination

Drawing on existing research [33–35] and taking into account the principles of science, operability, and accessibility, we first determined the dimensions of each system and then chose specific indicators of each dimension. The coupling and coordination system between the three systems is shown in Figure 1.

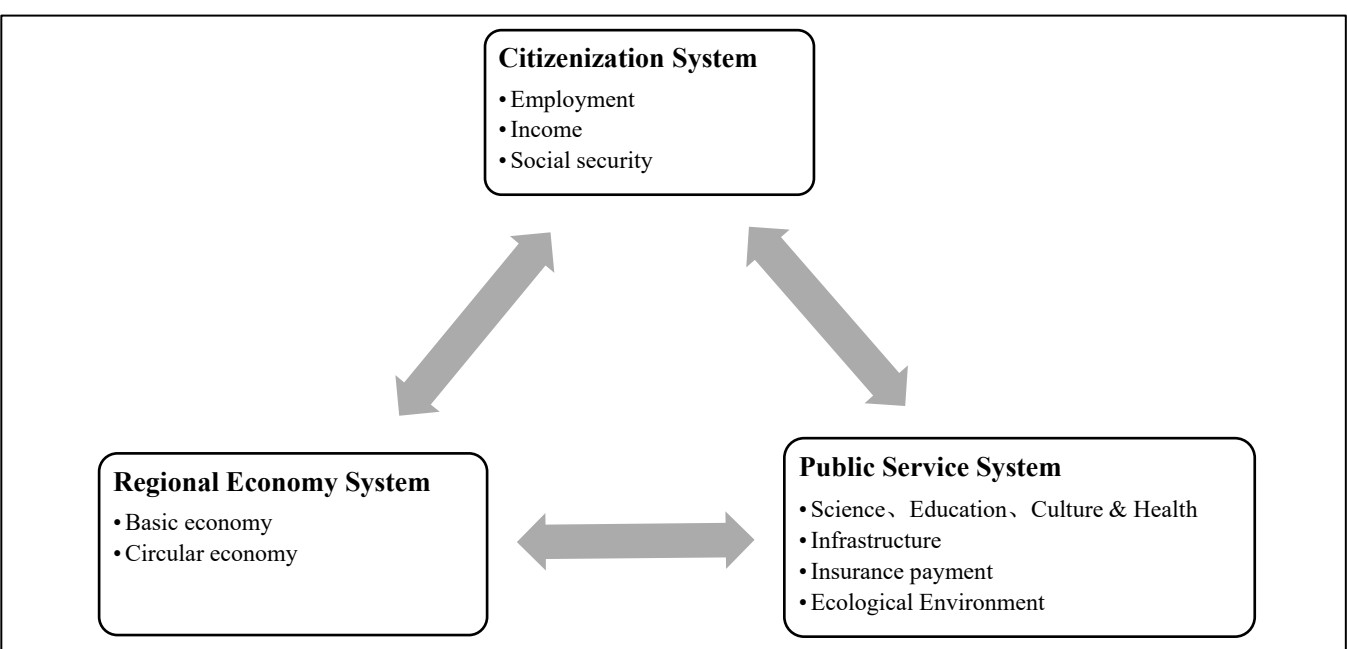

**Figure 1.** "Citizenization, Regional Economy, Public Service" index system.

Referring to the previous literature [33,36,37], we measured citizenization through three dimensions: employment, income, and social security, which were represented by eight indicators in detail. Each indicator's definition is shown in Table 2. First, the employment dimension includes: employed or not, work units, and employment status. This paper recoded the twelve categories of work units used in the CMDS into two categories: formal unit and informal unit. The former included public institutions, state-owned and state-controlled enterprises, collective enterprises, joint-stock/joint ventures, private enterprises, Hong Kong/Macao/Taiwan-owned enterprises, wholly foreign-owned enterprises, and Sino-foreign joint ventures, while the latter included individual industrial and commercial households, associations/private organizations, others, and no work unit. Second, the income dimension included relative individual monthly income, monthly household income per person, and purchasing power. We divided personal monthly income by the average wage of urban workers to get the relative personal monthly income and to set the result to be one if it is larger than one. The logic for this setting was that personal income exceeding the average wage of urban workers could be considered as having completed the citizenization in personal income. Third, the dimension of social security included the place of payment of medical insurance and the type of insurance. Both the place and the type of the medical insurance premium payment significantly impacted the medical security of rural-to-urban migrants in the inflow locale, since it was difficult to get compensation if the premium was paid in non-inflow locales. Moreover, the insurance coverage degree of different insurance types varied dramatically.

**Table 2.** The definition of the indicators of the citizenization index system of rural-to-urban migrants.

| Dimension | Indicators | Definition |
|---|---|---|
| Employment Citizenization | Employed or not | In employment = 1;<br>others = 0 |
| | Work units | Formal unit = 1;<br>informal unit = 0 |
| | Employment status | Employers, or employees with regular employers = 1;<br>self-employed, employees without regular employers,<br>or others = 0 |
| Income Citizenization | Individual monthly income | Calculated from the "Personal monthly income/average wage of urban workers" |
| | Monthly household income per person | Calculated from the "Monthly household income/number of people living with the household" |
| Social Security Citizenization | Purchasing power | Calculated from the "Household expenditure—housing expenditure" |
| | The place of payment of medical insurance | Pay the medical insurance premium payment in the inflow locales = 1;<br>others = 0 |
| | The type of insurance | Basic medical insurance for urban workers, or free medical service = 4;<br>basic medical insurance for urban residents = 3;<br>basic medical insurance for rural = 2;<br>new cooperative medical care system insurance = 1;<br>no insurance = 0 |

In terms of regional economy, the selection of indicators should not only consider the economic development itself but also the cost of the ecological environment and energy efficiency. Therefore, based on the basic economic dimension, this paper added a circular economy dimension to explore the impact of a green and low-carbon economy on the coordinated and sustainable development of the system. First, the basic economic dimension included six indicators: per capita GDP, the proportion of secondary industry in GDP, the proportion of tertiary industry in GDP, the proportion of fiscal revenue in GDP, the proportion of total import and export in GDP, and the total retail sales of consumer goods per capita. Second, the circular economy dimension included three indicators: the growth rate of total energy consumption, the growth rate of change in energy consumption per unit of GDP, and the growth rate of power consumption per unit of GDP. Among them, the total energy consumption refers to the total amount of various energy consumptions consumed in a country by various industries and households in a certain period of time. The growth rate of change of total energy consumption is the growth rate of change in year $t$ relative to year $t - 1$. Energy consumption per unit of GDP refers to the total amount of energy consumed by the country to produce one unit of GDP in the period $t$. The growth rate of energy consumption per unit of GDP is the growth rate of change in year $t$ relative to year $t - 1$. Power consumption per unit of GDP refers to the total amount of power consumed by the country to produce one unit of GDP in the period $t$. The growth rate of power consumption per unit of GDP is the growth rate of change in year $t$ relative to year $t - 1$. All three indicators were available directly from the China Energy Statistics Yearbook. Today, the energy shortage causes obvious constraints on China's social and economic development. Therefore, energy sustainability and cost-benefit provides essential security for sustainable economic growth.

In terms of public services, the scope of basic public services generally included education, medical care, housing security, social security, and culture and sports, as well as infrastructure fields such as transportation and communication that were closely related

to the living environment [37]. This paper drew on the existing research on the public service system and introduced three dimensions of science/education/culture/health, infrastructure, and premium payment. Furthermore, China pays great attention to the ecological economy and promotes green development. A good ecological environment is an important foundation for sustainable economic development. Therefore, this paper added the fourth dimension, the ecological environment dimension, and selected three measurable indicators: parks' green space area per capita, the harmless treatment rate of domestic waste, and the urban sewage treatment rate.

### 3.3. Measurement Method of Coupling Coordination Degree of "Citizenization, Regional Economy, Public Service" System

Then, we applied the entropy method to determine each indicator's weight within the respective system. Characteristics of information entropy can be used to measure the degree of dispersion of an index. The greater the index's degree of dispersion, the greater its impact on the overall evaluation and, thus, its weight. The entropy weight method is an objective assignment method that depends on the discreteness of the data itself. It is used to comprehensively score samples in combination with multiple indicators to achieve comparison between samples. This method avoids the subjectivity of artificial weighting and is more scientific. Specific steps are as follows:

First, to avoid the influence of dimensions, the indicators should be standardized. At the same time, to avoid generating 0 values, we adjusted the normalized equation so that the minimum value is not less than 0.1 [34]. When the indicator is a positive indicator, choose Equation (1) for standardization, and when the indicator is a negative indicator, choose Equation (2) for standardization.

$$x'_{ij} = (1 - 0.9) + \frac{0.9 * (x_{ij} - \min(\sum x_j))}{\max(\sum x_j) - \min(\sum x_j)} \tag{1}$$

$$x'_{ij} = (1 - 0.9) + \frac{0.9 * (\max(\sum x_j) - x_{ij})}{\max(\sum x_j) - \min(\sum x_j)} \tag{2}$$

Second, calculate the entropy value $p$ for each indicator. $P$ means the relative weight of the $i$th province under the $j$th indicator. See Equation (3) for more details.

$$p_{ij} = \frac{x'_{ij}}{\sum\limits_{i=1}^{n} x'_{ij}} \tag{3}$$

Third, calculate the entropy weight E of the indicator. The E value represents the information entropy of the $j$th indicator. The greater the information entropy, the smaller the amount of existing information. See Equation (4) for more details.

$$E_j = -\frac{1}{\ln n} \sum\limits_{i=1}^{n} p_{ij} \ln p_{ij} \tag{4}$$

Finally, the entropy weight of the indicator is obtained. See Equation (5) for more details. According to the entropy weight method, the relative weight of each index in the "Citizenization, Regional Economy, Public Service" System is obtained. See Table 3 for more details.

$$W_j = -\frac{1 - E_j}{k - \sum E_j} \tag{5}$$

**Table 3.** "Citizenization, regional economy, public service" system, indicator system, and weight in 2018.

| System | Dimension | Indicator | Weight |
|---|---|---|---|
| Citizenization System | Employment | Employed or not | 0.10 |
| | | Work units | 0.15 |
| | | Employment status | 0.11 |
| | Income | Individual monthly income | 0.09 |
| | | Monthly household income per person | 0.21 |
| | | Purchasing power | 0.08 |
| | Social security | The place of payment of medical insurance | 0.11 |
| | | The type of insurance | 0.14 |
| Regional Economy System | Basic economy | Per capita GDP (CNY 10,000) | 0.18 |
| | | The proportion of secondary industry in GDP (%) | 0.05 |
| | | The proportion of tertiary industry in GDP (%) | 0.14 |
| | | The proportion of fiscal revenue in GDP (%) | 0.07 |
| | | The proportion of total import and export in GDP (%) | 0.26 |
| | | The total retail sales of consumer goods per capita (CNY 10,000) | 0.19 |
| | Circulareconomy | The growth rate of total energy consumption (%) | 0.03 |
| | | The growth rate of change in energy consumption per unit of GDP (%) | 0.03 |
| | | The growth rate of power consumption per unit of GDP (%) | 0.06 |
| Public Service System | Science, Education, Culture, health | The proportion of education spending in public budget spending (%) | 0.07 |
| | | The prorpotion of science and technology spending in public budget spending (%) | 0.10 |
| | | Number of beds in medical and health institutions owned by 10,000 people (pieces per 10,000 people) | 0.05 |
| | | Number of books in public libraries per capita (volumes/person) | 0.09 |
| | Infrastructure | Urban road area per capita (square meters/person) | 0.03 |
| | | The number of buses and trams owned by 10,000 people (amount/10,000 people) | 0.10 |
| | | Length of urban water supply pipeline per capita (square meters/person) | 0.11 |
| | Insurance payment | The proportion of urban employees participating in the basic old-age insurance (%) | 0.10 |
| | | The proportion of basic medical insurance for urban employees (%) | 0.12 |
| | | Unemployment insurance participation (%) | 0.12 |
| | Ecological environment | Parks' green space area per capita ($m^2$/per person) | 0.05 |
| | | The harmless treatment rate of domestic waste (%) | 0.03 |
| | | Urban sewage treatment rate (%) | 0.03 |

After weights are determined, we used the standardized value of each indicator to calculate the weighted sum score of each dimension and each system. For each indicator, the indicator's minimum value was converted to 0, the maximum value was converted to 1, and all other values were converted to decimals between 0 and 1. It is worth noting that, when measuring the level of citizenization alone, in order to display the comparison clearly we processed the indicator on a percentage scale so that the urbanization indicator was within the range of 0–100.



After the scores of the three systems at the national level and by province were obtained, the coupling and coordination degree analysis between the three systems could be carried out [38,39]. The model involved three indexes: the coupling degree (denoted by C), the comprehensive development value (denoted by T), and the coupling coordination degree (denoted by D). Coupling degree C measures the correlation relationship between systems, representing the strength of their interaction; coordination index T represents the overall level of systems; coupling coordination degree D refers to the benign coupling degree between interactive systems. Given n $\geq$ 2 systems, $U_i \geq 0$ represents each system score. The calculation equation of coupling degree is as shown in Equation (6):

$$C(U_1, U_2, \ldots U_n) = n \times \left[ \frac{U_1 U_2 \ldots U_n}{(U_1 + U_2 + \ldots + U_n)^n} \right]^{\frac{1}{n}} \tag{6}$$

In Equation (6), n = 3, C represents the coupling and coordination relationship between systems, where U1 is the citizenization system of rural-to-urban migrants, U2 is the regional economic system, and U3 is the public service system. C is the coupling degree, C $\in$ (0, 1]. The closer C is to 1, the closer the development levels of the three systems are and the higher the coupling degree is. The degree of coupling describes the degree of interaction between systems but does not indicate whether the interaction occurs at a higher level or a lower level of system development. Thus, further measuring the D value of the coupling coordination degree is necessary. We combined the coordination status with the development level between the systems to make up for the defect of the coupling degree C.

The coupling coordination degree D value is as shown in Equation (7):

$$D = \sqrt{C * T} \tag{7}$$

In the above equation, T is the comprehensive development value that represents the comprehensive development level of the citizenization system, the regional economic system, and the public service system, which can be expressed as shown in Equation (8):

$$T = \beta_1 U_1 + \beta_2 U_2 + \beta_3 U_3 + \ldots \ldots + \beta_n U_n \tag{8}$$

β represents the weight. Considering that the three systems have equal importance, this paper makes $\beta_1 = \beta_2 = \beta_3 = 1/3$. Referring to the coordination degree classification principles of the literature, we categorized the coupling coordination degree of "citizenization, regional economy, public service" system into five categories [40]. See Table 4 for details.

**Table 4.** Coordination level division.

| Coupling Coordination | <0.6 | 0.6~0.7 | 0.7~0.8 | 0.8~0.9 | 0.9~1.0 |
|---|---|---|---|---|---|
| Coordination Level | Low | Medium-Low | Medium | Medium-High | High |

## 4. Coupling and Coordination Analysis

### 4.1. The Overall Level of Citizenization of Rural-to-Urban Migrants

The overall level of citizenization of rural-to-urban migrants was relatively low, and the dimensions of income and social security were shortcomings. In 2018, the citizenization score of rural-to-urban migrants was 50.26 and there was still room for substantial improvement. The score in the employment dimension was relatively high, reaching 59.45, characterized by a high employment ratio but low employment quality. The employment proportion of rural-to-urban migrants was 85.92%, however most were engaged in low-level manual labor jobs in cities. Regarding the nature of work units, 37.5% of the employed rural-to-urban migrants were self-employed and 9.4% did not have any work unit. Regarding employment status, only 57.75% were business owners or employees with formal contracts with their employers. See Table 5 for details.

**Table 5.** Citizenization scores of all indicators of China's rural-to-urban migrants in 2018.

| Dimension | Secondary Indicators | Score |
|---|---|---|
| Employment Citizenization | Employed or not | 85.92 |
| | Work units | 44.37 |
| | Employment status | 57.75 |
| Income Citizenization | Individual monthly income | 60.93 |
| | Monthly household income per person | 40.61 |
| Social Security Citizenization | Purchasing power | 40.30 |
| | The place of payment of medical insurance | 38.42 |
| | The type of insurance | 51.43 |

In contrast, the income and social security levels were relatively low at 45.62 and 45.70, respectively, and they could be important breakthrough points for promoting the citizenization of rural-to-urban migrants. In terms of income, the relative personal income was higher than the household income and purchasing power. In 2018, the average personal monthly income of employed rural-to-urban migrants was CNY 5549, with a corresponding score of 60.93. However, the household income and purchasing power scores were only 40.61 and 40.30, respectively. Regarding social security, the proportion of premium payments in inflow locales was only 38.4%, because the medical insurance system's problems such as the continuity, circulation, and settlement in different places had not been effectively solved. At the same time, the total proportion of free medical care and basic medical insurance for urban workers, both with higher health security, was only 31.3%.

### 4.2. The Development Trend of Circular Economy

The circular economy in the eastern regions had a more prosperous trend than it did in the other regions. The circular economy was not only an effective way to realize environmental protection and green economic growth but also an important choice for transforming the economic development mode. Figure 2 shows the growth rates of total energy consumption, energy consumption per CNY 10,000 of GDP, and power consumption per CNY 10,000 of GDP in eastern, central, western, and northeastern China from 2016 to 2020. Figure 2a shows the growth rate of total energy consumption in the three major regions from 2016 to 2020. It can be seen that, compared with other regions, the growth rate of total energy consumption in the eastern region had turned from positive to negative by 2020, while other regions were still positive. In other words, energy consumption in the eastern region had begun to decrease in 2020, while total energy consumption in other regions was still increasing compared with previous years. Figure 2b shows the growth rate of change in energy consumption per unit of GDP in the three regions from 2016 to 2020. The energy consumption per unit of GDP across all four regions experienced negative growth in 2016, but the absolute value of the negative growth rate was becoming smaller yearly, indicating that the room for decline was gradually shrinking, and the growth rate in the northeast had turned positive in 2020. The reduction rate of energy consumption in the central and western regions was also getting slower and slower, and only the eastern region still maintained a relatively high reduction rate of energy consumption per unit of GDP. Figure 2c shows the growth rate of power consumption per unit of GDP in the three major regions from 2016 to 2020. The growth rates of power consumption per unit of GDP were negative in all four regions in 2016. However, by 2020, only the east remained negative, while the other three regions turned positive. In other words, the power consumption per unit of GDP in the eastern region had shown a decreasing trend, while the power consumption per unit of GDP in the other regions had increased. Overall, it showed an upward trend nationwide. Compared with the other three regions, the eastern region

had the best performance. The total energy consumption growth rate turned positive to negative in 2020, and both energy consumption per unit of GDP and power consumption per unit of GDP had been decreasing year by year, indicating steady progress in green energy development, energy saving, and consumption reduction. The central and the northeast regions followed, and the western region had the worst performance. The total energy consumption growth rate in the western region was constantly staying at a relatively high level. Energy consumption per unit of GDP showed an upward trend, representing the poor performance of the circular economy.

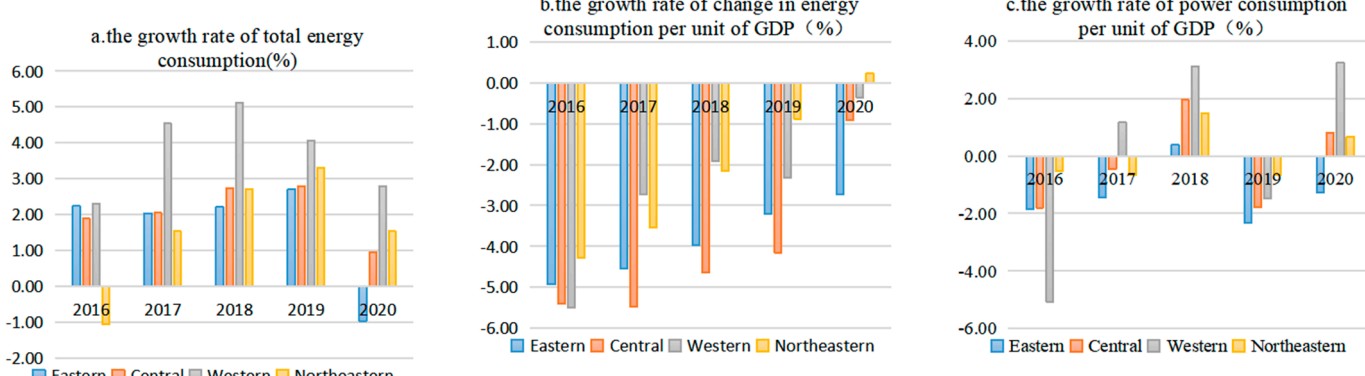

**Figure 2.** Changes in the main indicators of circular economy in the four regions of China from 2016 to 2020.

### 4.3. The Development Level of Region

The development level of each system varied significantly from region to region, and the overall trend was "high in the east and low in the west". In the spatial distribution pattern of the three systems, there were noticeable regional differences in the "citizenization, regional economy, public service" system, as shown in Figure 3. Figure 3a–c show the distribution of citizenization level, economic development level, and public services level by province in 2018. The darker the color, the higher the score. More detailed data are presented in Table 6. We found that economically developed areas such as Beijing, Shanghai, Zhejiang, Jiangsu, Guangdong, and other eastern coastal provinces enjoyed higher scores in the three systems. In contrast, the underdeveloped western parts such as Guizhou, Gansu, Yunnan, Qinghai, and Tibet had lower scores. Overall, the spatial characteristics of China's "citizenization, regional economy, public service" system showed a gradient distribution pattern that gradually decreased from east to west.

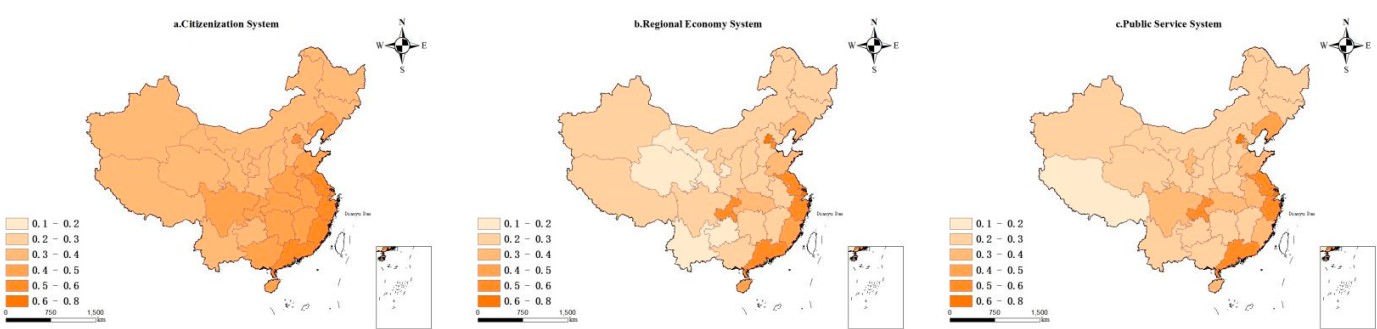

**Figure 3.** Spatial distributions of the citizenization system, the regional economy system, and the public service system in China by province.

**Table 6.** Relevant parameter results of the coupling coordination model by province in 2018.

| Province | Citizenization | Regional Economy | Public Service | C | T | D | Coordination Level |
|---|---|---|---|---|---|---|---|
| Mainland China | 0.5026 | 0.3757 | 0.3570 | 0.9884 | 0.4118 | 0.6380 | |
| Beijing | 0.5580 | 0.7617 | 0.7167 | 0.9912 | 0.6788 | 0.8203 | Medium-High |
| Shanghai | 0.5873 | 0.7643 | 0.6319 | 0.9938 | 0.6611 | 0.8106 | |
| Zhejiang | 0.5273 | 0.5593 | 0.5837 | 0.9991 | 0.5568 | 0.7458 | |
| Jiangsu | 0.5469 | 0.5466 | 0.5125 | 0.9995 | 0.5353 | 0.7315 | Medium |
| Guangdong | 0.5564 | 0.5065 | 0.5320 | 0.9993 | 0.5316 | 0.7289 | |
| Chongqing | 0.4651 | 0.5058 | 0.5391 | 0.9982 | 0.5033 | 0.7088 | |
| Tianjin | 0.4423 | 0.5080 | 0.4593 | 0.9983 | 0.4698 | 0.6849 | |
| Fujian | 0.5450 | 0.4245 | 0.3736 | 0.9876 | 0.4477 | 0.6649 | Medium-Low |
| Liaoning | 0.4315 | 0.3565 | 0.4643 | 0.9939 | 0.4174 | 0.6441 | |
| Hainan | 0.4269 | 0.4651 | 0.3345 | 0.9906 | 0.4088 | 0.6364 | |
| Shandong | 0.4243 | 0.3655 | 0.3963 | 0.9981 | 0.3954 | 0.6282 | |
| Hubei | 0.4179 | 0.3001 | 0.3321 | 0.9903 | 0.3501 | 0.5888 | |
| Anhui | 0.4357 | 0.2596 | 0.3188 | 0.9773 | 0.3381 | 0.5748 | |
| Sichuan | 0.4447 | 0.2578 | 0.3081 | 0.9740 | 0.3369 | 0.5728 | |
| Jilin | 0.3710 | 0.2782 | 0.2919 | 0.9919 | 0.3137 | 0.5578 | |
| Hunan | 0.4152 | 0.2629 | 0.2737 | 0.9780 | 0.3173 | 0.5570 | |
| Jiangxi | 0.4319 | 0.2292 | 0.2979 | 0.9665 | 0.3197 | 0.5558 | |
| Inner Mongolia | 0.3474 | 0.2979 | 0.2795 | 0.9958 | 0.3083 | 0.5541 | |
| Ningxia | 0.3581 | 0.2221 | 0.3511 | 0.9773 | 0.3104 | 0.5508 | |
| Shaanxi | 0.3353 | 0.2729 | 0.2836 | 0.9959 | 0.2973 | 0.5441 | |
| Henan | 0.4415 | 0.2298 | 0.2545 | 0.9578 | 0.3086 | 0.5436 | Low |
| Hebei | 0.3989 | 0.2433 | 0.2601 | 0.9753 | 0.3008 | 0.5416 | |
| Shanxi | 0.3299 | 0.2547 | 0.2839 | 0.9944 | 0.2895 | 0.5366 | |
| Heilongjiang | 0.3508 | 0.2503 | 0.2538 | 0.9875 | 0.2850 | 0.5305 | |
| Guangxi | 0.4173 | 0.2248 | 0.2340 | 0.9588 | 0.2920 | 0.5292 | |
| Xinjiang | 0.3783 | 0.2085 | 0.2738 | 0.9708 | 0.2869 | 0.5277 | |
| Guizhou | 0.3326 | 0.1781 | 0.2869 | 0.9671 | 0.2659 | 0.5071 | |
| Gansu | 0.3674 | 0.1852 | 0.2260 | 0.9581 | 0.2595 | 0.4987 | |
| Yunnan | 0.3500 | 0.1883 | 0.2196 | 0.9646 | 0.2526 | 0.4936 | |
| Qinghai | 0.3212 | 0.1931 | 0.2209 | 0.9764 | 0.2451 | 0.4892 | |
| Tibet | 0.3447 | 0.2222 | 0.1529 | 0.9465 | 0.2399 | 0.4765 | |

*4.4. The Degree of Coupling and Coordination*

The degree of coupling was high and the degree of coordination was low, while the degree of coupling and coordination was at a low level as a whole. Through the coupling coordination model, we calculated the coupling degree C, the comprehensive development value T, and the coupling coordination degree D of the "citizenization, regional economy, public service" system in 2018, as shown in Table 6. From the perspective of coupling degree C, the degree among the three systems in each province was higher than 0.9, which meant a strong interaction and interdependence between the systems. In terms of the comprehensive development value T, the national level was relatively low, at 0.41. However, it differentiated significantly across provinces. Among them, the comprehensive development value of Beijing was the highest, yet merely being 0.68, while the comprehensive development value of Tibet was the lowest, which was 0.24. In terms of the coupling coordination degree D, its national level was medium-low, being 0.64. According to the interval of the coupling coordination degree, 31 provinces could be classified into four categories: medium-high (0.8, 0.9], medium (0.7, 0.8], medium-low (0.6, 0.7], and low (0.6 and below). Most provinces were in medium-low or low level categories, and no province had a higher degree of coupling coordination than 0.9.

*4.5. The Degree of Coupling and Coordination of Region*

The more developed the region, the higher the coupling and coordination degree. As shown in Figure 4, the provinces with relatively high coupling coordination degrees D were located in the more developed eastern region, including Beijing and Shanghai

with medium-high level D, and Zhejiang, Jiangsu, Guangdong, and Chongqing with medium level D; the reason being that local governments needed to afford the social cost of citizenization. Therefore, provinces with a strong economy could provide better financial support for the citizenization of rural-to-urban migrants and better supplies of public services, which was conducive to promoting the benign interaction of the three systems. On the contrary, underdeveloped provinces such as Guizhou, Gansu, Yunnan, Qinghai, and Tibet had difficulty covering the cost of citizenization, and the supply capacity of public services was also limited, which hindered the benign interaction between the three systems, resulting in a lower coupling coordination degree.

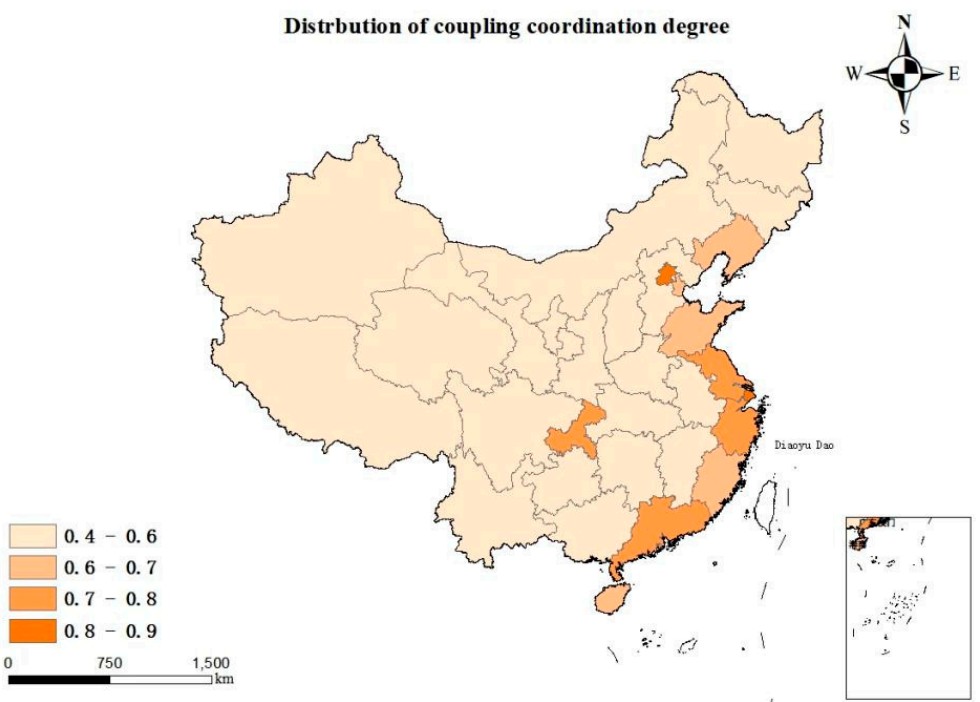

**Figure 4.** Distribution of coupling coordination degree of "citizenization, regional economy, and public services" system by province in China (2018).

### 4.6. Characteristics of Coordinated Development in the Region

The public service supply and economic development level in most provinces lagged behind their citizenization level. Public service supply and economic development lagged behind citizenization in 24 out of 31 provinces, of which Tibet, Henan, Guangxi, and Gansu held the largest gap. See Figure 5 for details. While the citizenization of these provinces was inadequate, their public service supply and regional economy scores were even lower. It implied that the citizenization progress in these provinces was more motivated by administrative directives than by economic development and the improvement of public service supply. The level of citizenization was not "the higher, the better". The one that matched the development levels of economy and public service is the best. Passive or excessive citizenization driven by the administrative directives from provincial governments might create new urban–rural contradictions or social conflicts, which are not conducive to sustainable development.

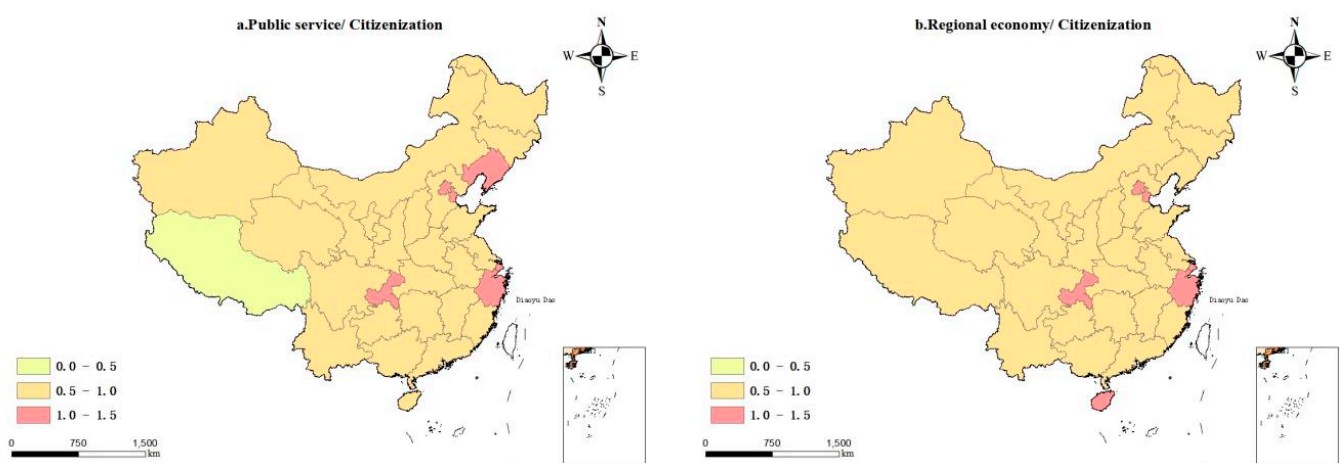

**Figure 5.** Distributions of the ratio of public service to citizenization, and the ratio of regional economy to citizenization by province in 2018.

The citizenization system in developed provinces was lagging behind the other two systems, showing an opposite characteristic compared with the underdeveloped ones. On the contrary, the citizenization in Beijing, Shanghai, Tianjin, Chongqing, and Zhejiang lagged behind the economic development and the public service supply. These five provinces belonged to the type of "citizenization lagging". These provinces' economy and public service supply capacities were sufficient to facilitate further citizenization. However, the governments tended to set up many administrative obstacles, hindering the process of citizenization. It was particularly evident in Beijing and Shanghai, two megacities. Impelled by the redline of population size during the "13th Five-Year Plan" period (2016–2020), migration policies in Beijing and Shanghai had been further tightened. As a result, both cities once experienced significant negative population growth during the period, and the process of citizenization fell behind the economy and the public services significantly. As a result, the ratios of the scores of the economic system to the citizenization system in Beijing and Shanghai were 1.37 and 1.30, respectively, while the ratios of the scores of the public service system to the citizenization system were 1.28 and 1.08, respectively, which were significantly higher than other provinces. It suggests that megacities need to remove institutional barriers so that rural-to-urban migrants can get more access to urban public services, as well as to promote the equalization of public services and to enhance citizenization level so that they can achieve a higher degree of coupling and coordination.

## 5. Research Conclusions and Policy Enlightenment

This paper uses the 2018 CMDS data and statistical yearbook data to establish measurement systems for the citizenization of rural-to-urban migrants, regional economic development, and public services, respectively. Based on this, we build up a coupling coordination model between the three systems. Then, we analyze the characteristics of the regional development status of the three systems, as well as the spatial coupling and coordination relationship between the systems, and explore the sustainable development path that adapts citizenization to regional economic development and public service supply.

There are two contributions of this paper. First, it is the first time that the citizenization system, regional economy, and public service system are put into the same analytical framework. It discusses the coupling coordination degree between the three systems and analyzes their coordination relationship, as well as their regional differences. This research helps to explore the effective promotion of citizenization from a macro perspective, and the relevant conclusions can provide a reference for regional policy formulation. Secondly, to fully reflect the requirements of sustainable development in the selection of indicators, this paper introduces elements of the circular economy into the economic system and elements of environmental protection into the public service system. These indicators make

the construction of the framework more reasonable and help to promote the high-quality development of the regional economy and the public service system while promoting citizenization.

The main research conclusions are the following:

First, the degree of coupling and coordination between the three systems is generally low. Most provinces are at a low level of coupling and coordination, and sustainable development is limited. The study found that the current coupling coordination between the three systems is at a medium-low level. Among the 31 provinces counted in this paper, 20 provinces are at a low level of coordination. Regional sustainable development needs to rely on a multi-dimensional and dynamic equilibrium, including population, economy, society, resources, ecology, etc. The current level of coupling and coordination between the three systems of citizenization, regional economy, and public services is low, which will be detrimental to the simultaneous advancement between regional sustainable development and new-type urbanization.

Second, the principal contradiction in the problem of coupling and coordination between systems is entirely different in developed and underdeveloped provinces. Therefore, it is necessary to seek sustainable development paths that adapt to local conditions. Although the degree of coupling and coordination among the three systems is generally low, there are some significant differences between provinces regarding their contradictory characteristics. Most provinces show that the level of economic development and public service is lagging behind citizenization, while some developed provinces represented by megacities show that their citizenization system is lagging behind. It is necessary to remove institutional barriers further and to improve the high-quality coordinated development of systems between citizenization, regional economy, and public services.

Third, the overall level of the citizenization of rural-to-urban migrants is relatively low and the dimensions of income and social security are shortcomings, which will have an adverse impact on the sustainable development of new-type urbanization. The citizenization of rural-to-urban migrants is the primary task of new-type urbanization. It is necessary to gradually ensure that rural-to-urban migrants have the same employment opportunity and environment as urban residents in the inflow locales and to improve their income and social security level, which will finally promote the sustainable development of new-type urbanization.

Fourth, the circular economy in the eastern regions has a more prosperous trend than it does in the other regions. As a new mode of economic development, the circular economy is an inevitable choice to achieve sustainable development. The growth rate of change in total energy consumption, energy consumption per unit of GDP, and power consumption per unit of GDP in the eastern region have all shown a downward trend, forming a relatively green and low-carbon development trend. The economic system has an inhibitory effect, which is not conducive to the region's sustainable development.

Fifth, the development level of the three systems is generally "high in the east and low in the west". The economically developed provinces in the eastern coastal areas are at a relatively high level in the "citizenization, regional economy, public service" system, while the provinces in the western regions are at a relatively low level and need to be further improved.

Based on the above research, this paper proposes that: aiming at sustainable development, we should comprehensively promote the coordinated development of citizenization, regional economy, and public services, as well as improve the degree of coupling and coordination between the systems. First of all, for some developed provinces where the development of citizenization is lagging behind, it is necessary to remove *hukou* barriers further and to promote that rural-to-urban migrants can enjoy more benefits from urban development, leading to a higher degree of coupling and coordination. Secondly, for areas with obvious passive citizenization characteristics, it is necessary to gradually expand the coverage of various public services for rural-to-urban migrants. Through the development of the two systems of the regional economy and the public services, the development of the

citizenization system will be driven and, eventually, a high level of coordination between the three will be realized. Finally, the development of a circular economy in the central and western regions lags behind that in the eastern region, which is one of the reasons for the low score of its regional economic system. While promoting citizenization, it is necessary to focus on developing a circular economy and on strengthening ecological environment construction. It is also necessary to guide the transformation of the industrial structure into one with high scientific and technological content and economic benefits, as well as to enhance the ability of regional sustainable development.

In future work, it can be explored from both researching objects and trends. On the one hand, in this study, we take provinces as the research unit, but future studies can take cities, even counties, as the unit, which will help the government formulate more targeted policies. On the other hand, the coupling and coordination degree of citizenization, regional economy, and public services is a process of dynamic change, and vertical tracking can give us deeper insight into trends, motivations, and mechanisms.

**Author Contributions:** Conceptualization, M.Q. and P.M.; methodology, M.Q. and P.M.; software, P.M., Y.W. and Y.S.; validation, M.Q.; formal analysis, P.M. and Y.W.; investigation, M.Q., P.M. and Y.W.; resources, P.M. and M.Q.; data curation, P.M., Y.W. and Y.S.; writing—original draft preparation, P.M. and M.Q.; writing—review and editing, M.Q., P.M., Y.W. and Y.S.; visualization, P.M., Y.W. and Y.S.; supervision, M.Q.; project administration, M.Q.; funding acquisition, M.Q. All authors have read and agreed to the published version of the manuscript.

**Funding:** This research was funded by the National Social Science Foundation of China, grant number: 18BRK005.

**Institutional Review Board Statement:** Not applicable.

**Informed Consent Statement:** Not applicable.

**Data Availability Statement:** All the data used in this study are public.

**Conflicts of Interest:** The authors declare no conflict of interest.

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
