# Peer review of "Research on the Coupling and Coordination of Systems of Citizenization, Regional Economy, and Public Service in China from the Perspective of Sustainable Development"

_sustainability, doi:10.3390/su141912916_

Round 1

Reviewer 1 Report

This paper used CMDS and statistical yearbook to built up three index systems, including the citizenization, economic development, and public services. Innovatively, authors took the circular economy into the economic system and environmental protection into the public service system. Methods are suitable. Conclusions are relatively accurate. Generally, this paper is suitable to be publish in Sustainability after major revisions.

Here are some suggestions that may be useful for you.

1. Introduction

Line 36: Please add reference based on your first sentence (The transfer of agricultural population to cities occurs all over the world).

3. Research Methods

Line 154-157: Please add a table that displays each indicator’s data source. Please add the study period of each indicator. Which year you used?

Figure 1: The resolution is too low, please provide a higher resolution image.

Table 2: The weight of two indicators (Monthly household income per person and The proportion of total import and export in gdp) are high, is this similar to other references. Please add some details about this. In addition, in this table, gdp should be “GDP”.

What’s more, the ecological environment only has three sub-indicators. Suggest you consider the air quality index, like the PM2.5 concentration.

Line 226-227: coupling and coordinated degree model, T doesn’t refer to the coordination index, based on (2), it referred to the comprehensive development value. Suggest you read some reference:

Page 9: https://doi.org/10.1007/s10668-021-01975-z

Page 4: https://doi.org/10.1016/j.jclepro.2019.05.027

Line 252: Table 3, why you divided the coordination index into five categories, is there any reference or method you used. Please add some details.

4. Coupling and coordination analysis

As for the title of 4.1, 4.2, 4.3, 4.4, 4.5, 4.6, 4.7 are they suitable to let the result as the title.

Figure 2: the resolution is too low. Please add a higher one.

Line 258: in this part, you used 0-100 to describe their score. However, in Figure 4 and Table 4, you used 0-1. I didn’t find some descriptions about how you transform it. Even it is obvious that you divide 100. However, in paper, you need add this sentence. Please modify it.

Figure 4&5&6: (1) The resolution is low, please provide a higher one. (2) Please provide the project information of this map. (3) The small picture, you need to redrawn, please add the Diaoyu Dao and South China Sea ten segment line. (This is important, you can refer to Figure 1 in this paper https://doi.org/10.3390/rs14030737).

Table 4: Many provinces’ C values equals 1.00, when used the equation of C, take Jiangsu as an example, the C value equals 0.9993, suggest you keep four decimal places.

5. Research conclusions and policy enlightenment

The dataset of citizenization index you used is from CMDS, and as you described in 3.1, you only used 2018, so why you calculate other years index in Figure 3. Or you just display three sub-indexes’ result?

Author Response

Dear reviewer,

Thank you very much for your comments and professional advice.These suggestions have helped us a lot.Under your suggestion, we have made up for some defects and got rid of a lot of mistakes. At the same time, some necessary content has been added to make the article more readable.We hope that our work can be improved again. Furthermore, we would like to show the details in document.

1.Introduction

Line 36: Please add reference based on your first sentence (The transfer of agricultural population to cities occurs all over the world).

Response from authors: We have added relevant references according to the suggestions.

3.Research Methods

Line 154-157: Please add a table that displays each indicator’s data source. Please add the study period of each indicator. Which year you used?

Response from authors: As suggested, we have supplemented and improved each indicator's data sources and years and added a table for data sources.( refer to 3.1 and Table 1)

Figure 1: The resolution is too low, please provide a higher resolution image.

Response from authors:We have replaced Figure 1 and provided a higher resolution image.

Table 2: The weight of two indicators (Monthly household income per person and The proportion of total import and export in gdp) are high, is this similar to other references. Please add some details about this. In addition, in this table, gdp should be “GDP”.

Response from authors: (1)we apply the entropy method to determine each indicator's weight within the respective system. Characteristics of information entropy can be used to measure the degree of dispersion of an index. The greater the index's degree of dispersion, the greater its impact on the overall evaluation, and, thus, its weight.The entropy weight method determines the weight of the evaluation indicator based on sample data, not on subjective interpretation.(2)Thank you for your correction, we have revised the GDP.

What’s more, the ecological environment only has three sub-indicators. Suggest you consider the air quality index, like the PM2.5 concentration.

Response from authors: Thank you for your valuable suggestions. When constructing the index system, we screened many indicators, including PM2.5, but we found that when we put PM2.5 into the system, it is roughly consistent with the existing results. Thus, PM2.5 was not included in the framework, and instead, we chose the typical indicators of greening, domestic waste, and sewage.

Line 226-227: coupling and coordinated degree model, T doesn’t refer to the coordination index, based on (2), it referred to the comprehensive development value. Suggest you read some reference:

Page 9: https://doi.org/10.1007/s10668-021-01975-z

Page 4: https://doi.org/10.1016/j.jclepro.2019.05.027

Response from authors: Thank you for your sharing. After carefully reading these two articles, we realized our lack of understanding and made corresponding modifications to the content of the coupling coordination model in the article.

Line 252: Table 3, why you divided the coordination index into five categories, is there any reference or method you used. Please add some details.

Response from authors:We drew on relevant literature to classify the coordination coupling indicator and supplemented the literature citations in the paper.(refer to line 313)

4.Coupling and coordination analysis

As for the title of 4.1, 4.2, 4.3, 4.4, 4.5, 4.6, 4.7 are they suitable to let the result as the title.

Response from authors: Thank you for your suggestion. We have revised as requested.

Figure 2: the resolution is too low. Please add a higher one.

Response from authors:To show the content more clearly, we replaced Figure 2 with a table 5.

Line 258: in this part, you used 0-100 to describe their score. However, in Figure 4 and Table 4, you used 0-1. I didn’t find some descriptions about how you transform it. Even it is obvious that you divide 100. However, in paper, you need add this sentence. Please modify it.

Response from authors:Thank you for your suggestion. We have made corresponding supplements in the paper. (refer to line 281-286) 

Figure 4&5&6: (1) The resolution is low, please provide a higher one. (2) Please provide the project information of this map. (3) The small picture, you need to redraw, please add the Diaoyu Dao and South China Sea ten segment line. (This is important, you can refer to Figure 1 in this paper https://doi.org/10.3390/rs14030737).

Response from authors:Thank you for your suggestion. We have made the replacement as requested. (refer to Figure 3&4&5)

Table 4: Many provinces’ C values equals 1.00, when used the equation of C, take Jiangsu as an example, the C value equals 0.9993, suggest you keep four decimal places.

Response from authors: Thank you for your suggestion. We have rounded to 4 decimal places as requested. (refer to Table 6)

5.Research conclusions and policy enlightenment

The dataset of citizenization index you used is from CMDS, and as you described in 3.1, you only used 2018, so why you calculate other years index in Figure 3. Or you just display three sub-indexes’ results?

Response from authors:Thank you for your suggestion. The data sources used in the paper were supplemented and improved. ï¼ˆ refer to 3.1 and Table 1)

Yours sincerely

23 September 2022

Reviewer 2 Report

General comments:

I have found the topic interesting. However, it is unclear what problem your paper wants to address. In other words, what do we learn from this paper that we do not know from the current literature? The contribution is not clear. The paper needs substantial improvement regarding these points. 

- In the introduction, several literatures cited are not convincing enough for bring up the problem. 

-The rationality of the structure and content for the paper needs to be further improved, as the position of the contributions needs to adjust, the conclusion part of the paper fails to extract the research conclusion.

Author Response

Dear reviewer,

Thank you very much for your comments and professional advice.These suggestions have helped us a lot. According to your suggestions,We have revised the framework and the structure of the paper, to make the article structure more reasonable and the content more readable,including revising the introduction, supplementing the literature, revising the conclusions and suggestions, summarizing the research contributions. We hope that our work can be improved again. Furthermore, we would like to show the details in document.

- In the introduction, several literatures cited are not convincing enough for bring up the problem. 

Response from authors: Thank you for your suggestion. we have added some literature and content in the literature review to make the article framework more complete . In addition, we have further revised the conclusions and suggestions,and summarized the research contributions.

-The rationality of the structure and content for the paper needs to be further improved, as the position of the contributions needs to adjust, the conclusion part of the paper fails to extract the research conclusion.

Response from authors: Thank you for your suggestion. We have further revised our conclusion and research contributions.

Yours sincerely

23 September 2022

Author Response

Dear reviewer,

Thank you very much for your comments and professional advice.These suggestions have helped us a lot. According to your suggestions, we have reorganized the article , including revising the introduction, supplementing the literature, revising the conclusions and suggestions, summarizing the research contributions.We hope that our work can be improved again. Furthermore, we would like to show the details in document.

1.The introduction should provide, briefly and clearly: problem definition, gaps in the literature, problems solution, study motivation, paper's aims & objectives, and significance and advantages of the research. When formulating the research/study goal, it would be worth writing what the scientific (cognitive) goal was and what was the practical (utilitarian) goal of the research.

Response from authors: Thank you for your suggestion. We have reorganized the content in the introduction.

2.Literature Review: it remains rather vague about the research/study goal and authors` concept (i.e., citizenization). Generally speaking, in this form, literature does not present previous international research related to the research area. What does previous research say about the topic? As a result, Literature Review must present logical RQs.

Response from authors: Thank you for your suggestion. We supplemented the literature review on the concept of citizenization and its relevant research (refer to line 83-86). In addition, the household registration problem involved in citizenization is unique to China, so in this part of the literature review, we showed more relevant research progress in China.

3.The Research Conclusions and Policy Enlightenment (description of results) must be linked to the scope of the research. Why are the results important or relevant to your readers? Do they add further evidence to a scientific consensus or disprove prior studies? Conclusions should be structured as theoretical contributions, practical implications, limitations, and future research perspectives. Please, explain how your study provides a new understanding or fresh insights into the problem. The discussion will always connect to the introduction through the research questions you posed and the literature you reviewed. Also, it should explain how your study has moved the reader's understanding of the research problem forward from where you left them at the end of the introduction, connected to current literature, which must be more extended to significant authors beyond Chinese authors. I hope these comments are helpful when revising the manuscript, and I wish you all the best for your future Research in this area.

Response from authors: Thank you for your suggestion. We have fully revised our conclusions and research contributions. We rearranged and reiterated it into four parts: theoretical contribution,  main conclusion, practical significance, and future research prospects.

Yours sincerely

23 September 2022

Reviewer 4 Report

Here are my comments on the paper, Research on the Coupling and Coordination of Systems of Citizenization, Regional Economy, and Public Service in China from the Perspective of Sustainable Development,  submitted to Sustainability

1. The abstract is quite long.  The authors should shorten the abstract to make it more succinct for the readers.

2. Line 41, had already exceeded 250 million. Although they have moved to cities, they have  should be rewritten as  exceeded 250 million. Although they  moved to cities, they have

3. Line 90 and 91, it is unclear what this means.  It needs to revised. 

4. LInes 134-140 appears to belong in the introduction section of the paper. It does not make much sense that it is in its current location. 

5. Figure 1 is quite difficult to read.  The authors would need to enlarge it. 

6. Lines 203-204 it is unclear what this means.  It needs to be revised. 

7.  Line 202, unit of GDP. Today, the energy shortage has caused obvious constraints on China's  should be rewritten as unit of GDP. Today, the energy shortage caused obvious constraints on China's

8. Line 216  there is mention of entropy. I suspect this is discussed elsewhere in the paper?

9. Under section 4.2, there is a discussion of the circular economy. There needs to be a discussion of the circular economy as a subset under the review of the literature section.  I suspect that some readers may not be fully familiar with the circular economy. 

10. Line 302 performance of circular economy.  should be rewritten as performance of the circular economy.

11. In 307 there was mention of the spatial distribution patterns. How was this done? 

12. Line 314 as Guizhou, Gansu, Yunnan, Qinghai and Tibet suffered lower scores. Overall, the  should be rewritten as as Guizhou, Gansu, Yunnan, Qinghai and Tibet had lower scores. Overall, the

13. Line 312 calculate should be calculated 

14. Figures 5 and 6 are quite difficult to read.  The authors need to enlarge these figures so the readers can be able to read them.

15. Under section 5, there is a discussion about the results of the circular economy.  It is not quite clear how the estimates were prepared for the circular economy.  There are numerous methods to calculate the circular economy, or there is no set methodology. The authors need to explain how these results were calculated carefully. 

16. How does this paper contribute to the various strands of the literature? 

17. Lines 451-458  This sentence is too long. You are being a bit too wordy.  LESS IS MORE IN GOOD WRITING.  Eliminate unnecessary words and make your writing as tight and concise as you can.

Author Response

Dear reviewer,

Thank you very much for your careful reading, and for your comments and professional advice. These suggestions have helped us a lot. And we have made up for many defects based on your advice. At the same time, we have added or reduced some content to optimize the framework and structure of the article. We hope that our work will be improved once again. Also, we want to show these details in the documentation.

  1. The abstract is quite long.  The authors should shorten the abstract to make it more succinct for the readers.

Response from authors: Thank you for your suggestion. We have simplified the abstract. 

  1. Line 41, had already exceeded 250 million. Although they have moved to cities, they have  should be rewritten as  exceeded 250 million. Although they  moved to cities, they have

Response from authors: Thank you for your correction. We have made corresponding changes.

  1. Line 90 and 91, it is unclear what this means.  It needs to revised. 

Response from authors: Thank you for your suggestion. We reiterated the content of this paragraph as follows: “Ali (2013) took Pakistan as a sample, and the results also verified that public service expenditure positively affects economic growth .”

  1. LInes 134-140 appears to belong in the introduction section of the paper. It does not make much sense that it is in its current location. 

Response from authors: Thank you for your suggestion, we added the corresponding content to the introduction. At the same time, we have added the corresponding literature on sustainable development and some content in the literature review part, to make this part serve as a further interpretation.

  1. Figure 1 is quite difficult to read.  The authors would need to enlarge it. 

Response from authors: We performed replacement and provided a higher resolution image

for Figure 1.

  1. Lines 203-204 it is unclear what this means.  It needs to be revised. 

Response from authors: Thank you for your suggestion. We reiterated the content of this paragraph as follows: “China pays great attention to the ecological economy and promotes green development. A good ecological environment is an important foundation for sustainable economic development.”

  1. Line 202, unit of GDP. Today, the energy shortage has caused obvious constraints on China's  should be rewritten as unit of GDP. Today, the energy shortage caused obvious constraints on China's

Response from authors: Thank you for your correction. We have made corresponding changes.

  1. Line 216  there is mention of entropy. I suspect this is discussed elsewhere in the paper?

Response from authors: Thank you for your suggestion. We have supplemented the concept and steps of the entropy weight method accordingly (refer to 3.3).

  1. Under section 4.2, there is a discussion of the circular economy. There needs to be a discussion of the circular economy as a subset under the review of the literature section.  I suspect that some readers may not be fully familiar with the circular economy. 

Response from authors: Thank you for your suggestion. We have sorted out relevant literature for the circular economy and included it in the literature review (refer to 2).

  1. Line 302 performance of circular economy.  should be rewritten as performance of the circular economy.

Response from authors: Thank you for your correction. We have made corresponding changes.

  1. In 307 there was mention of the spatial distribution patterns. How was this done? 

Response from authors: Thank you for your correction. Their scores are calculated by the entropy weight method (refer to 3.3), and each province has a corresponding score for the citizenization system, regional economic system, and public service system. In figure3, we depicted it as a map, and more detailed data can be found in table6, Columns 2 to 4 (refer to table6).

  1. Line 314 as Guizhou, Gansu, Yunnan, Qinghai and Tibet suffered lower scores. Overall, the  should be rewritten as as Guizhou, Gansu, Yunnan, Qinghai and Tibet had lower scores. Overall, the

Response from authors: Thank you for your correction. We have made corresponding changes.

  1. Line 312 calculate should be calculated 

Response from authors:Thank you for your correction. Our data are presented in the table6 (below), while making this point explicitly in the paper (refer to 378).

Province

Citizenization

Regional

Economy

Public

Service

Beijing

0.5580

0.7617

0.7167

Shanghai

0.5873

0.7643

0.6319

Zhejiang

0.5273

0.5593

0.5837

Jiangsu

0.5469

0.5466

0.5125

Guangdong

0.5564

0.5065

0.5320

  1. Figures 5 and 6 are quite difficult to read.  The authors need to enlarge these figures so the readers can be able to read them.

Response from authors: Thank you for your suggestion. We have made corresponding adjustments to Figures 5 and 6 to express more clearly (refer to figure 4&5).

  1. Under section 5, there is a discussion about the results of the circular economy.  It is not quite clear how the estimates were prepared for the circular economy.  There are numerous methods to calculate the circular economy, or there is no set methodology. The authors need to explain how these results were calculated carefully. 

Response from authors: Thank you for your suggestion. We have added the definitions, calculation methods, and data sources from the three indicators in the circular economy (refer to line 227-237).

  1. How does this paper contribute to the various strands of the literature? 

Response from authors: Thank you for your correction. We have made relevant supplements in the literature review section and have presented the contributions of this paper in conclusion in Section 5. “There are two contributions this paper has made. First, it is the first time that the citizenization system, regional economy, and public service system are put into the same analytical framework. It discusses the coupling coordination degree between the three systems and analyze their coordination relationship, as well as their regional differences. This research helps to explore the effective promotion of citizenization from a macro perspective, and the relevant conclusions can provide a reference for regional policy formulation. Secondly, to fully reflect the requirements of sustainable development in the selection of indicators, this paper introduces elements of the circular economy into the economic system and elements of environmental protection into the public service system. These indicators make the construction of the framework more reasonable and help promote the high-quality development of the regional economy and public service system while promoting citizenization.”

  1. Lines 451-458  This sentence is too long. You are being a bit too wordy.  LESS IS MORE IN GOOD WRITING.  Eliminate unnecessary words and make your writing as tight and concise as you can.

Response from authors: Thank you for your correction. The following revision has been made. “It is necessary to remove hukou barriers further, promote the rural-to-urban migrants can enjoy more benefits from urban development, leading to a higher degree of coupling and coordination.”

Yours sincerely

23 September 2022

Round 2

Reviewer 1 Report

Authors have well addressed all my comments. So, I recommend this manuscript to be published in Sustainability journal.

Author Response

Dear reviewer,

Thank you for carefully examining my manuscript. I hope to have the opportunity to get your guidance again in the future. Your careful work and thoughtful suggestions have brought me great help and helped me make significant improvements to this paper. Under your guidance, we have made up for many defects of paper. Thank you once again sincerely for your efforts.

Reviewer 3 Report

Accept in present form

Author Response

Dear reviewer,

Thank you for carefully examining my manuscript. I hope to have the opportunity to get your guidance again in the future. Your careful work and thoughtful suggestions have brought me great help and helped me make significant improvements to this paper. Under your guidance, we have optimized the structure of the article to make the content richer and the theme more clear, Thank you once again sincerely for your efforts.

Reviewer 4 Report

Here are my comments on the paper Research on the Coupling and Coordination of Systems of Citizenization, Regional Economy, and Public Service in China from the Perspective of Sustainable Development submitted to Sustainability

1. Table 2 is quite difficult to read.  The authors need to improve table 2 in the paper.

2. Carefully ensure that the review of the literature and the empirical results are written in the past tense because the work has already been completed.

3. Lines 173-174 "in order to"  should be replaced with "to";  this comment can also apply to line 255

3a.  Insert the before specific in line 254

4.  The title for Table 1 should be Data Sources

5.  Line 369 there should be a comma after Qinghai; Line 397 there should be a comma after Guangdong; Line 402  there should be a comma after Qinghai; Line 412 there should be a comma after Guangxi; Line 426 there should be a comma after Chongqing

6.Line 451 There are two contributions this paper has made-- should be rewritten as There are two contributions of this paper. 

7. Line 530 wang should be capitalized for W

8.  The references do not conform to the style required by Sustainability.  The titles of journals is not spelled out but abbreviated. Also, in the references, the last of the multiple authors should be in ampersand not the word and.  Again the authors need to check the style guidelines for the references and correct as necessary. 

9.  Lines 334-335  "The circular economy in the eastern regions has a more prosperous trend than it does  in the other regions."  How was this shown?  Need to clarify this statement.

10.  It is more customary to use the word "equation" not "formula" in scientific writing.  The authors should use the word equation. 

11.  Line 276 do not use it's but spell it out as it is

Author Response

Dear reviewer,

Thank you very much for your comments and professional advice again.These suggestions have helped us a lot.Under your suggestion, we have made up for some defects and got rid of a lot of mistakes. At the same time, some necessary content has been added to make the article more readable.We hope that our work can be improved again. Furthermore, we would like to show the details in document.

  1. Table 2 is quite difficult to read. The authors need to improve table 2 in the paper.

Response from authors: Thank you for your suggestion. ï¼ˆ1)Table 2 shows the definition of the indicators of the citizenization system. We have modified the content and expression form of Table 2 to make it clearer. (2)Referring to previous literature, we measure citizenization through three dimensions: employment, income, and social security, which are represented by eight indicators in detail. These eight indicators are all derived from the question items in "the China Migrants Dynamic Survey (CMDS)". Among them, the definition of indicators is set according to the options of the question in CMDS. For example,employment status was obtained by the question of "which one is your current employment status?" in the CMDS, where we defined "employers, or employees with regular employers" as 1 and "self-employed, employees without regular employers, or others" as 0.

  1. Carefully ensure that the review of the literature and the empirical results are written in the past tense because the work has already been completed.

Response from authors: Thank you for your suggestion. We checked and revised the tense of the paper. 

  1. Lines 173-174 "in order to"  should be replaced with "to";  this comment can also apply to line 255

3a. Insert the before specific in line 254

Response from authors: Thank you. We changed it as you suggested.

  1. The title for Table 1 should be Data Sources

Response from authors: Thank you for your correction. We have made corresponding changes.

  1. Line 369 there should be a comma after Qinghai; Line 397 there should be a comma after Guangdong; Line 402  there should be a comma after Qinghai; Line 412 there should be a comma after Guangxi; Line 426 there should be a comma after Chongqing

Response from authors: Thank you for your correction. We have made corresponding changes.

  1. Line 451 There are two contributions this paper has made-- should be rewritten as There are two contributions of this paper. 

Response from authors: Thank you for your correction. We have made corresponding changes.

  1. Line 530 wang should be capitalized for W

Response from authors: Thank you for your correction. We have made corresponding changes.

  1. The references do not conform to the style required by Sustainability.  The titles of journals is not spelled out but abbreviated. Also, in the references, the last of the multiple authors should be in ampersand not the word and.  Again the authors need to check the style guidelines for the references and correct as necessary. 

Response from authors: Thank you for your correction. We have made corresponding changes.

  1. Lines 334-335  "The circular economy in the eastern regions has a more prosperous trend than it does in the other regions."  How was this shown?  Need to clarify this statement.

Response from authors: Thank you for your suggestion. We have added relevant content to provide a clearer explanation of our point.

  1. It is more customary to use the word "equation" not "formula" in scientific writing.  The authors should use the word equation. 

Response from authors: Thank you for your correction. We have made corresponding changes.

  1. Line 276 do not use it's but spell it out as it is

Response from authors: Thank you for your correction. We have made corresponding changes.

Yours sincerely

2 October 2022
